# Parametric Design of Porous Structure and Optimal Porosity Gradient Distribution Based on Root-Shaped Implants

**DOI:** 10.3390/ma17051137

**Published:** 2024-02-29

**Authors:** Lijian Liu, Shaobo Ma, Yongkang Zhang, Shouxiao Zhu, Shuxuan Wu, Guang Liu, Guang Yang

**Affiliations:** School of Mechanical Engineering, Hebei University of Science and Technology, Shijiazhuang 050018, China

**Keywords:** parametric design, porous structure, porosity gradient distribution, root-shaped implants

## Abstract

Porous structures can reduce the elastic modulus of implants, decrease stress shielding, and avoid bone loss in the alveolar bone and aseptic loosening of implants; however, there is a mismatch between yield strength and elastic modulus as well as biocompatibility problems. This study aimed to investigate the parametric design method of porous root-shaped implants to reduce the stress-shielding effect and improve the biocompatibility and long-term stability and effectiveness of the implants. Firstly, the porous structure part was parametrically designed, and the control of porosity gradient distribution was achieved by using the fitting relationship between porosity and bias and the position function of bias. In addition, the optimal distribution law of the porous structure was explored through mechanical and hydrodynamic analyses of the porous structure. Finally, the biomechanical properties were verified using simulated implant–bone tissue interface micromotion values. The results showed that the effects of marginal and central porosity on yield strength were linear, with the elastic modulus decreasing from 18.9 to 10.1 GPa in the range of 20–35% for marginal porosity, with a maximum decrease of 46.6%; the changes in the central porosity had a more consistent effect on the elastic modulus, ranging from 18.9 to 15.3 GPa in the range of 50–90%, with a maximum downward shift of 19%. The central porosity had a more significant effect on permeability, ranging from 1.9 × 10^−7^ m^2^ to 4.9 × 10^−7^ m^2^ with a maximum enhancement of 61.2%. The analysis showed that the edge structure had a more substantial impact on the mechanical properties. The central structure could increase the permeability more effectively. Hence, the porous structure with reasonable gradient distribution had a better match between mechanical properties and flow properties. The simulated implantation results showed that the porous implant with proper porosity gradient distribution had better biomechanical properties.

## 1. Introduction

The aging of the population in today’s society and the gradual improvement of living standards have led to an increasing demand for the restoration of missing teeth. For patients, missing teeth will not only have a particular impact on daily eating but will also cause impediments to speech and communication and even cause changes in the appearance of the facial and oral areas, so it is necessary to replace the missing tooth roots with dental implants [1]. Currently, threaded cylindrical implants are mainly used in clinical practice, but there is a morphological mismatch between them and the extraction sockets; some scholars have found that personalized root-shaped implants are more advantageous than conventional implants in terms of improving the distribution of stress [2,3]. The root-analogue implant is a type of implant that mimics a natural tooth root and is shaped to match the human socket highly. Due to the fact that the shape of the implant corresponds to the shape of the root of the extracted tooth or the extraction socket, the root-analog implant has an advantage over the conventional screw implant in terms of the distribution of stress at the interfacial interface with the bone tissue [4]. Because personalized implants mimic the morphology of the natural tooth root and do not require the preparation of an implant socket at the time of implantation, damage to the surrounding bone and soft tissues can be minimized during implant surgery. However, it has been suggested that root-shaped implants have insufficient stability at the interface with the alveolar bone at the initial stage of implantation and that macroscopic features need to be added to their surfaces [5].

The medical material Ti6Al4V has good biocompatibility and is a conventional material for the preparation of orthopedic implants. However, the elastic modulus of Ti6Al4V on the market reaches 110 GPa, whereas the elastic modulus of human cortical bone is 3–30 GPa, and that of cancellous bone is 1–3 GPa. If implants are prepared using titanium alloys, the high hardness of the implant leads to a non-physiological distribution of the loads, the so-called stress shielding [6,7]; the resulting uneven stress distribution can lead to bone loss, aseptic loosening, and fatigue damage due to overloading of the skeleton [8]. Applying a porous structure not only effectively reduces the modulus of elasticity and reduces stress shielding, but it also facilitates tissue regeneration and allows oxygen and nutrients to diffuse [9].

In recent years, the design of implants with predominantly porous structures has gained particular interest in personalized implant applications, as they can be effectively used to replace broken or damaged bone [10,11,12,13]. The main approaches to the design of porous structures include the design of regular porous structures based on primary cells [14] and triply-periodic minimal surfaces (TPMS) [15,16,17] as well as the realization of irregularly porous structures using image processing [18] and computer programs [19,20,21]. Among them, a triply-periodic minimal surfaces (TPMS) is a highly small-value surface with a complex 3D topological space structure; its internal structure is interconnected, smooth-surfaced, and has the characteristics of high specific surface area and high porosity, which is an ideal porous structure for orthopedic implant design. Compared to straight-edged or sharply curved structures with geometrically primitive shapes such as cylinders and cubes, TPMS structures have a surface curvature similar to that of natural bones with smooth surfaces, which can provide a better biomorphic environment for cellular activities (e.g., cell attachment, infiltration, migration, and proliferation). Secondly, due to the curvature of the structure, the TPMS structure can also reduce stress concentration and thus increase mechanical strength. Rajagopalan et al. [22] tested the activity of TPMS structures on cells by culturing ATDC cells on the scaffolds, and examination of the results after one day showed that the cells were in an active state and well-distributed on the surface of the TPMS structures; this proves that the TPMS structures have a natural biophilic design and provides sufficient evidence for their use as viable bone tissue analogues. However, the porous Ti alloys commonly used in clinical practice need to be more high-strength, low-modulus matching, i.e., the strength of porous alloys cannot meet the implantation requirements at high porosity. Related studies have also shown that cell adhesion and cell proliferation do not have the same need for pore size. Sun et al. (2018) developed a finite element model for bone implants and provided a generic approach for designing patient-specific implants with gradient modulus distributions [23]. Matena, Chang, et al. [24,25] showed in vitro that cell adhesion and differentiation are more favorable when the pore size is 188 μm. In comparison, the pore size is more advantageous for cell proliferation when the pore size is 313–390 μm.

The TPMS-G model has a high specific surface area, similar to the pore structure of cancellous bone. Different surface structures of TPMS have their own unique implicit function expressions. When designing the structure, the size of the porosity and the size of the corresponding porous units can be regulated by changing different parameters of the implicit function. Shi et al. (2019) parametrically designed porous structures based on triply-periodic minimal surfaces (TPMS), which have an excellent osseointegration effect [26]. In this study, we chose the commonly used spiral icosahedral structure (gyroid, G) as the pore structure design model in the TPMS model and changed the size and porosity of the pore unit by modulating the parameters of the implicit function of the TPMS-G model, to realize the modeling of homogeneous pore structure with different pore diameters and the gradient pore structure of the G unit, and simulated and analyzed the static mechanics and permeability of the structure, and then optimized the optimal unit structure. A gradient structure with an average porosity of 55% was then designed based on the optimal unit structure, and the static mechanics and permeability were simulated and analyzed to optimize the optimal porosity; finally, the optimized gradient structure with optimal porosity was placed into the oral environment, and further analyses were carried out in terms of the stress–strain distribution between the implant and the bone tissue as well as the micromovement at the implant–bone interface, which led to the optimal porosity; the optimal porous structure was further analyzed. Further analyses were conducted to derive the optimal gradient structure model suitable for oral implantation. This paper aims to study the porous root-shaped implant, to improve the degree of individuality of the implant, and to design a mechanically well-matched and suitably permeable porous structure based on the influence of the pore porosity distribution pattern on the mechanical and permeability properties. The final result is a better match between the implant and the human body.

## 2. Structural Design of Gyroid Porous Bone Scaffolds

Triply Periodic Minimal Surfaces (TPMS) are periodic in three independent directions in space, and the mean curvature of each point on the surface is 0. Therefore, they have the advantage of having a variety of geometries, and parametric mathematical models can be constructed to describe them, as shown in Figure 1. Figure 1h shows the porous structure with different porosity gradient distributions, which is parametrically designed based on the triply-periodic minimal surface of Figure 1d. Figure 1a–c are the schematic diagrams of the placement of the porous structure and the pre-processing of the finite element analysis.

In non-parametric designs, most of the structures are similar, and only minor changes can be made [27]; by modifying the parameters of the TPMS implicit function expression, precise control of the aperture size and shape can be achieved [28,29]. Gyroid is a kind of triply-periodic minimal surface characterized by easily tunable structural parameters. The implicit function expression *f*_(*G*__)_ for a G-type surface is shown in Equation (1).
(1)$$f_{\left(G\right)}=\cos\left(\frac{2\pi }{T}x\right)\sin\left(\frac{2\pi }{T}y\right)+\cos\left(\frac{2\pi }{T}y\right)\sin\left(\frac{2\pi }{T}z\right)+\cos\left(\frac{2\pi }{T}z\right)\sin\left(\frac{2\pi }{T}x\right)=C$$
where *x*, *y*, *z* are spatial coordinates; *ω* = *2π*/*T*; *T* is the surface period, and a change in the period size in a given region will cause a change in the size of the cell structure; *C* is the surface bias, which can control the shape of the surface and the surface area of the corresponding TPMS structure.

Assuming that the surface in Equation (1) is the interface between porous and solid regions, and defining *f*_(*G*)_ < *C* as the pore region and *f*_(*G*__)_ > *C* as the stable region, the volume of the pore and concrete areas, and thus the cell porosity, can be changed by varying the size of *C*. The TPMS surfaces may be self-intersecting, and the G-type surfaces are set from 0 to 1.0 to prevent the formation of porous cell solids or connecting holes with a threshold value of C that is either too large or too small. In order to avoid the formation of solid pore units or connecting pore structures if the threshold *C* is too large or too small, we set the range of the *C* value of the G-type surface to 0–0.9 and take the values in steps of 0.1, build the corresponding 3D model to calculate the cell porosity, and then fit the mathematical correspondence between the porosity of the hole units of the G-type TPMS curved surfaces and the threshold *C*. As shown in Figure 1a, according to the fitting results, it can be found that there is a roughly linear relationship between the porosity of G-type structure and the bias *C*. The fitted relationship function is shown in Equation (2).
(2)P=0.33+0.5C*P* is the porosity of the porous structure, *C* is the bias of the porous structure.

To construct a positional relationship for porosity, it is sufficient to obtain a positional relationship for the bias, which ultimately gives the porous structure a bias of
(3)$$C=a*r^{n}+b$$
(4)$$r=\sqrt{x^{2}+y^{2}}$$

In the formula, *x* and *y* represent the positions in two directions. Three parameters, *a*, *b*, and *n*, can achieve the gradient regulation of the porosity of the porous structure and control the rate of change of porosity; *b* is the value of bias at the center point, and *n* can change the rule of change of porosity. In this way, the surface can obtain utterly different bias amounts at different positions, thus constructing the positional relationship of the porosity. The structural average porosity is calculated as
(5)$$\bar{P}=\frac{V}{\pi r^{2}H}=\frac{\int_{0}^{r}2\pi HP(r)dr}{\pi r^{2}H}=\frac{nP_{i}+2P_{e}}{n+2}$$*V* is the volume of the porous portion of the porous scaffold, obtained by integrating over *r* with the porosity function *P(r)*. When the values of *Pi*, *Pe*, and $\bar{P}$ are given, the value of *n* can be calculated. When *Pi* at the center *r*_0_ and *Pe* at the edge *r* are known, the corresponding *Ci*, *Ce* can be obtained by bringing them into Equation (2).
(6)$$\frac{Pi-0.33}{0.5}=C_{i}$$
(7)$$\frac{Pe-0.33}{0.5}=C_{e}$$

By associating Equation (6) as a function of center porosity, edge porosity, and the corresponding bias, it is represented with Equation (3).
(8)$$C_{i}=a*r_{0}+b$$
(9)$$C_{e}=a*r+b$$
where *r*_0_ is the center position, and *r*_0_ = 0, the joint Equations (5)–(9) can find the unknown:$$a=\frac{Pe-Pi}{0.5}*r^{-2\frac{Pe-\bar{P}}{\bar{P}-Pe}}$$
$$b=\frac{Pi-0.33}{0.5}$$
$$n=2\frac{Pe-\bar{P}}{\bar{P}-Pe}$$

## 3. Modelling and Finite Element Simulation

### 3.1. Modelling

In this paper, the root shape of the tooth was used as the study object, and the pore size should be greater than 300 μm, considering the need to promote the generation of new bone and capillaries [30]. Jetté et al., after analyzing the pore size through static testing (2018), suggested that the porosity should be between 40% and 70% when designing porous structures [13]. The porosity of TPMS structures should be manageable, as excessive porosity can significantly reduce the mechanical properties of TPMS structures [31]. Esen and Bor recommended using interconnected macroporous structures with porosity more significant than 55% for dental applications to improve inward cell growth [32]. Huang et al. comparatively analyzed 36% and 55% of porosity implants based on micro-CT images and numerical prediction and found that only 55% had more pronounced inward bone growth at week six [33]. To further explore the influence of porosity distribution on the mechanical properties of the porous structure within a suitable porosity range, the trigonometric expressions of the absorbent structure were calculated by using the built-in RegionPlot3D function and Export function of Wolfram Mathematica 12.0.1 (US) software. The porous structures with average porosity of 55%, edge porosity of 20%, 30%, 40%, and 50%, and center porosity of 60%, 70%, 80%, and 90% were generated, which were combined into 16 porous structures with different gradient porosity distribution variations. As shown in Figure 1h, it is a porous structure with a constant porosity of 60% at the center and varying porosity from low to high at the edges.The use of Materialise Mimics 21.0(BEL) on the original tooth root shape and the alveolar bone model extraction set the spacing between the cortical bone and cancellous bone for 2 mm.

### 3.2. Compression Simulation Settings

The STL model generated by Mathematica is a triangular patch mesh and cannot be analyzed by finite elements. Figure 1f shows that Hypermesh envelops the surface mesh model to form a tetrahedral mesh model that can be used for finite element analysis, and the tetrahedral mesh has the characteristics of fast generation speed and slightly lower calculation accuracy, which is verified by mesh independence to meet the requirements. According to the force of human teeth bites, the boundary conditions of the porous structure are approximated by applying −4 mm displacement load downward on the upper-end face of the structure and fixing the lower-end face of the model to observe the structural, mechanical properties and the changing law of the gradient distribution of different porosities. When the modulus of elasticity is reduced to be comparable with the modulus of elasticity of human natural bone, it can better reduce the stress-shielding effect. The compression yield strength (σy) is the stress value when the bone scaffold produces 0.2% residual deformation. It is the lowest stress for the bone scaffold to have permanent plastic deformation. The bone scaffold bearing force exceeding the compression yield strength (σy) will cause the bone scaffold to deform plastically and fail, an indicator of the bearing capacity of the bone scaffold. The greater the compression yield strength, the greater the impact force the bone scaffold can withstand in an accident such as a fall or a drop, and the less likely it is to fail.

SLM (Selective Laser Melting) is a metal additive manufacturing technology. During SLM printing, metal powder is sprayed onto a work platform in a cascading manner, layer by layer, and then a laser beam is used to melt specific areas of the metal powder to build the desired part layer by layer. This technique allows complex-shaped metal parts to be manufactured without the cutting and casting processes of traditional machining. A Renishaw AM250 (Gloucestershire, London, UK) was chosen to print sixteen 10 × 10 × 10 mm^3^ gradient porous structural samples. Firstly, it is necessary to prepare the construction platform and load titanium alloy powder, export the porous structure model to Stl format and import it into the software that comes with the machine, set up the placement position and angle, and then set up the printer parameters, as shown in Table 1, and then separate the porous structure from the substrate by wire-cutting after printing. A CMT5105 (Eden Prairie, MN, USA)-type universal testing machine was selected to test the printed porous structure samples to carry out the relevant experiments; the constant compression rate was set to 1 mm/min, and the downward stroke to 4 mm to reach the maximum range of 100 kN or fracture to stop the loading, and the final load and displacement data were collected to obtain the modulus of elasticity and yield strength. Finally, the experimental results of the porous structure are analyzed in comparison with the simulation results.

### 3.3. Fluid Simulation Settings

A fluid domain model of a gradient porous structure can be obtained by taking negative values for the model’s bias using bias functions with different porosities; the boundary conditions are set in the Fluid Flow (Fluent) module in ANSY Workbench 2023 R1(US). The boundary conditions for the velocity inlet and outlet of the model are given. In order to keep the model in a laminar flow state, the inlet velocity is given as 0.01 m/s. The rest of the surfaces of the simulation model are set as wall boundary conditions, and the wall is defined as a non-slip wall, which is used to constrain the fluid and solid regions [34].

Specific surface area and permeability in hydrodynamic analysis are two important indicators of the permeability performance of porous structures. In general, the larger the specific surface area, the larger the internal pores of the porous structure, the more conducive to cell adhesion and proliferation, and the specific surface area can reflect the permeability of the porous structure and under a certain porosity, the porous structure with a smaller specific surface area possesses a higher permeability. The wall shear stress can characterize the growth and attachment of osteoblasts within the bone scaffold; however, too much or too little wall shear stress will result in a wall unsuitable for cell growth and attachment. Permeability determines, to a certain extent, the capability for nutrient transport and waste metabolism in the porous structure, and appropriate permeability can promote tissue regeneration and benefit the scaffold implantation rate [35,36]. However, increasing the permeability will reduce the mechanical strength of the porous structure, so it is crucial to balance the permeability and mechanical properties of the absorbent structure.

### 3.4. Biomechanical Properties Simulation Setup

After reading the spiral CT tomographic data in DICOM format using Mimics 21.0(BEL) it was converted into the internal standard format of the software, and the original cross-sectional map was set to be extracted by region splitting and region growing for the extraction of the tooth region and the generation of the tooth mask. Based on the missing teeth mirror model mask, the 3D reconstruction of the missing teeth mirror model was performed. The model was optimized using Geomagic Studio (US) to improve its quality and accuracy. Figure 2 shows the mandible and lateral incisor models extracted from the human head model, with intercepted mandible segments assembled with the lateral incisors, after which the model was used as a basis for finite element pre-processing.

All interfaces between the implant and bone tissue were set to have completed osseointegration, and the contact interface between the implant and the restorative crown was set to be bound (bonded). All solid bodies were defined as linear, homogeneous, continuous, isotropic, and elastic materials; the modulus of elasticity and Poisson’s ratio of the various materials are shown in Table 2. The implant and the crown were placed in the same position. A load of 100 N was applied over the crown to replace the occlusal force, and the applied load was inclined from the palatal to the buccal side at 45° to the long axis of the implant [37]. The cells on the proximal and distal mesial sides of the jaws were set to remain stationary.

## 4. Results and Discussion

### 4.1. Mechanical Properties Analysis

In the compression simulation results, comparing the gyroid porous structure with different porosity gradient distributions, taking the edge porosity as 30% and the internal porosity as 80%, the comparative analyses, as shown in Figure 3 and Figure 4, show that the stress distributions are generally the same. The principal stresses are distributed at the intersection position of the adjacent cycles. The stress on the vertical strut is more significant than that on the horizontal strut, and the vertical strut plays a vital role in bearing force, so the mechanical properties of the gyroid porous structure can be improved by thickening the vertical strut and optimizing the shape of the vertical strut.

The simulation results of elastic modulus and yield strength are shown in Table 3; the elastic modulus of the natural bone tissue ranges from 0.76–19 GPa, the elastic modulus of all the gradient porous structures tested meet the requirements, and the relationship curves between the mechanical properties of the bone scaffolds and the porosity are plotted according to the data in the table. The elastic modulus of the bone scaffolds is most significant at 60% Pi and 20% Pe with an elasticity modulus of 18.92 GPa and smallest at 90% Pi and 50% Pe with an elasticity modulus of 4.78 GPa, which is 74.7% lower compared with the former. The modulus of elasticity is the smallest at 90% Pi and 50% Pe, which is 4.78 GPa, a decrease of 74.7% compared with the former. In addition, it can be observed from Figure 5 and Figure 6 that when the central porosity is kept constant, the most significant effect on the reduction of elastic modulus is observed in the range of 20–35% for the edge porosity, while the change of the central porosity has a lesser effect on the elastic modulus; the yield strength is maximum at 784.32 MPa for Pi of 60% and Pe of 20%, and the strain at the time of yielding is 0.25; in the case of Pi of 90% and Pe of 50%, the yield strength is minimum at 219.46 MPa, which is 72% lower compared with the former, and the strain at the time of yielding is 0.037. Further, 90% Pe is 50% yield strength, which is the smallest at 219.46 MPa, compared with the former reduced by 72% to reach the yield strain of 0.037. Edge porosity of 20–50% on the yield strength of the law is more consistent; with the edge of the porosity of the more apparent linear relationship, the edge of the porosity of 30% can be more significant to reduce the phenomenon of stress shielding.

From the calculation of the experimental simulation, it can be concluded that the experimental data of the gradient porous structure are all lower than the simulation data. However, they are all within the acceptable range. From Table 2, it can be seen that the experimental data of the mechanical parameters of the gradient porous structure drop more significantly in the low-porosity structure; on the one hand, the reason is that there are internal pores close to the boundary position in the preparation of the standard porosity structure, and on the other hand, it is because of the errors in the SLM molding process, and the process of laser melting, in which the more significant temperature gradient leads to the high residual stresses and imbalance of the microstructure.

The deformation patterns of the three porous structures recorded during compression are shown in Figure 7. It can be observed that the porous structures with low porosity at the edges and low porosity at the center all show a shear band at 45° to the compression direction at the stage of disruption. During compressive deformation, the dense metal usually fractures in a brittle manner, with the cracks oriented at 45° to the applied stress, and the 45° orientation of the cracks, as shown in Figure 7a, elucidates the brittle deformation behavior of porous Ti6Al4V samples. However, the brittle fracture of porous Ti6Al4V implants is due to the presence of dense walls rather than pores [38]. As the central porosity increases, the central porosity and edge porosity become more and more apparent, and the failure mode of the porous structure is gradually complicated; in the medium-porosity porous structure, the structure presents a section of vertical fracture surface after the edge fracture on the upper surface, until the overall failure of the sample; the fracture of the porous structure with a high porosity gradient occurs on the upper surface. Then, the fracture damage gradually occurs along the two sides, and the which ultimately leads to structural separation. The last part of the structure is separated from the structure in an inverted triangular shape.

As can be seen from the comparison in Table 4, the porous structure with gradient distribution of porosity has a significant advantage in matching the yield strength and modulus of elasticity. Although some porous structures have similar mechanical properties, the designed porous structure with a loose center and dense exterior is still advantageous in fluid-flow properties in the subsequent study.

### 4.2. Fluid Dynamics Analysis

In terms of the fluid stress score, as shown in Figure 8, the gradient porous structure with an average porosity of 55% and comparing it with the homogeneous porous structure with a porosity of 55%, it is found that the central porosity of the gradient porous structure is larger than the edge porosity. The main structure has less obstruction to fluid flow. The fluid mainly flows in the central structure, whereas the edge structure obstructs fluid due to decreased porosity. However, the flow velocity distribution of the gradient porous structure is greater. However, the flow distribution of the gradient porous structure is more biocompatible, the fast flow rate in the center is suitable for transporting blood oxygen and nutrients, and the slow flow rate in the periphery is more conducive to the proliferation and attachment of bone cells. A related study shows that the wall shear stress between 0.00005–0.1 MPa is more conducive to the proliferation of bone cells. As shown in Figure 9, in the area extracted in the range of 0.00005–0.1 MPa, the homogeneous porous structure accounted for 66.4%. In comparison, the gradient porous structure accounted for 75.8%, which is a 9.4% increase in the area conducive to the growth and proliferation of cells compared with the homogeneous porous structure.

Figure 10 and Figure 11 reveal that the maximum permeability of 6.4 × 10^−7^ m^2^ is observed for 90% central porosity and 50% external porosity, and the minimum permeability of 1.9 × 10^−7^ m^2^ is observed for 60% central porosity and 20% external porosity. Comparative analysis of Figure 10 and Figure 11 shows that the change in the central porosity has a more pronounced effect on the permeability of the porous structure than the change in the marginal porosity. The effect of change in central porosity on the permeability of the porous structure is more pronounced than that of change in marginal porosity, which can be seen in Figure 11. In contrast, the increase in central porosity does not have much effect on the permeability of the porous structure in the case of higher marginal porosity.

### 4.3. Biomechanical Analysis

Compared to homogeneous porous implants, gradient porous implants have lower cortical bone equivalent stress values and higher cancellous bone equivalent stress values in the implant; the maximum stress level at the implant–bone boundary affects the biological response, including bone resorption and remodeling [42]; according to bone mechanics [43], when the stress value of bone tissue is lower than 2 MPa, the bone tissue will produce wasteful bone resorption; when the stress value is between 2 and 60 MPa, the bone tissue maintains a normal bone state and is in an active state of bone building, which can promote the growth of bone tissue. When the stress value is between 2~60 MPa, the bone tissue maintains the normal bone state and is in the active state of bone plasticity, which can promote the growth of bone tissue. Comparative analysis of Figure 12a,c,e,g shows that the difference in stress value of bone tissue around the homogeneous porous structure is significant, with a maximum stress value of 48.377 MPa for cortical bone and a minimum stress value of 1.229 MPa for cancellous bone, which will cause waste of bone resorption; the difference in stress value of bone tissue around the gradient porous implant is slight, with the maximum stress value of 25.997 MPa for cortical bone and 3.497 MPa for cancellous bone, which can maintain the normal bone state; a gradient porous implant can promote bone growth. The gradient porous implant improves the stress transfer to the surrounding bone tissue more obviously than the homogeneous porous structure implant. It can increase the stress value of cancellous bone to prevent it from generating wasteful bone resorption so that the bone tissue can maintain the normal bone state. Wolff’s law shows that when the equivalent strain on the bone tissue is less than 100 microstrain, the bone tissue will undergo wasteful resorption, and when the equivalent strain is more than 3000 microstrain, the bone tissue will undergo wasteful resorption. Strain greater than 3000 microstrain will be pathological overload, so the equivalent strain between 100 microstrain and 3000 microstrain is suitable for bone tissue reconstruction [44]. Comparative analysis of Figure 12b,d,f,h shows that the difference between the aerodynamic values of the homogeneous porous structure around the bone tissue between the cortical bone and cancellous bone is 4152 for the maximum microstrain of the cortical bone and 922 for the cancellous bone, which makes it easy for pathological overloading to occur in the cortical bone; in contrast, the difference in the microstrain values of the bone tissue around the gradient porous implant is slight. The change is relatively smooth, with the maximum microstrain of the cortical bone being 2133 and that of the cancellous bone 2400; both are within a reasonable range, which can ensure the normal state of bone and promote the growth of bone.

## 5. Conclusions

In order to reduce the stress-shielding effect and promote the long-term stability of dental implants, a TPMS-G porous structure was selected for the design of the spongy part of the dental implants to explore the impact of changes in porosity, to obtain implants with a better match between mechanical and biological properties. The following main conclusions were drawn through finite element simulation.

The mechanical properties of the gradient porous structure have a more significant effect on the elastic modulus than the edge porosity, and the effect is more meaningful when the edge porosity is in the range of 20–35%.The central porosity has a more significant effect on permeability reduction, but the change of central porosity has almost no impact on permeability when the marginal porosity is more important.The biomechanical properties of the gradient porous structure and the homogeneous porous structure can be analyzed to conclude that the stress-shielding effect of the gradient porous structure is more evident than that of the homogeneous porous structure, which can significantly reduce the stress concentration.

This type of gradient porous structure has good mechanical matching and suitable flow properties, and its use in root-shaped implants facilitates inward bone growth. It can improve the long-term stability of root-shaped implants. However, this study still has some limitations; there is still some work that needs to be further investigated. In this paper, only finite element means were used to simulate the implantation of root-shaped porous implants, which lacked reliable clinical data research. In future studies, combining the clinical data of root-shaped porous implants will provide a better guarantee for practical medical applications.

## Figures and Tables

**Figure 1 materials-17-01137-f001:**
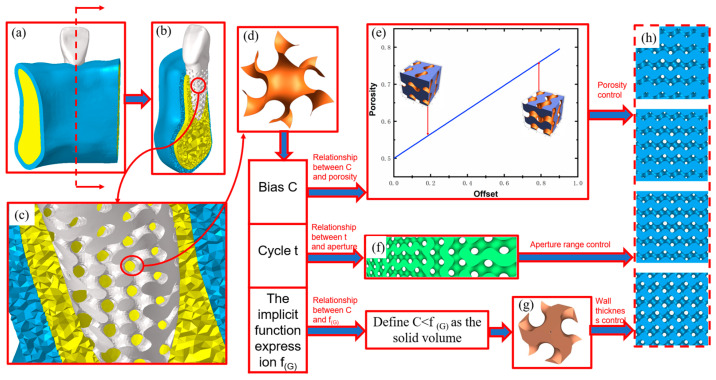
Parametric design of root-shaped porous implants. (**a**) Schematic of rooted implant assembly, (**b**) Diagram of the internal structure of the root implant, (**c**) The porous structure of the implant put in the picture, (**d**) Gyroid surface, (**e**) Fitting curve of bias and porosity, (**f**) Influence of period T on the porous structure, (**g**) Gyroid surface with bias C introduced to give thickness, (**h**) Schematic diagram of gradient porous structure with constant central porosity.

**Figure 2 materials-17-01137-f002:**
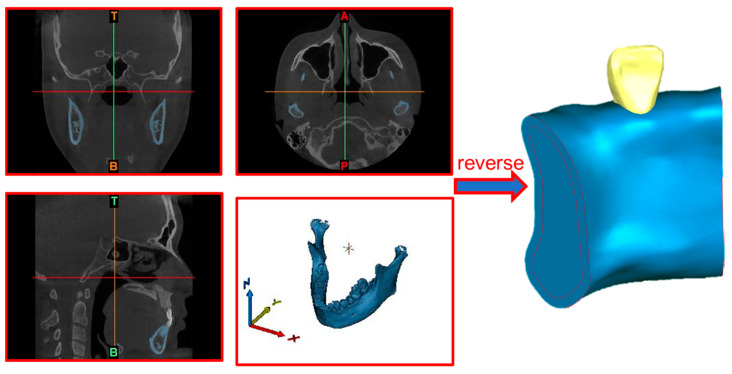
Simulation model reconstruction.

**Figure 3 materials-17-01137-f003:**
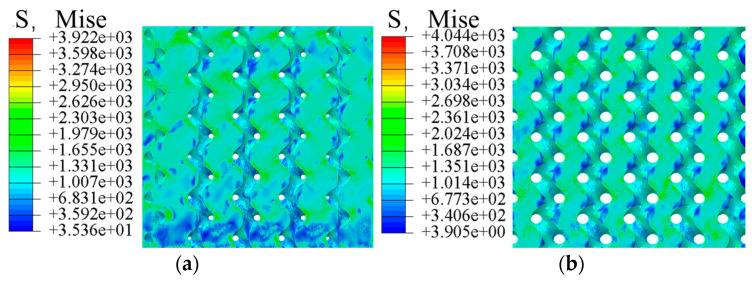
Homogeneous gradient porous structure strain: (**a**) Gradient porous structure; (**b**) Homogeneous porous structure.

**Figure 4 materials-17-01137-f004:**
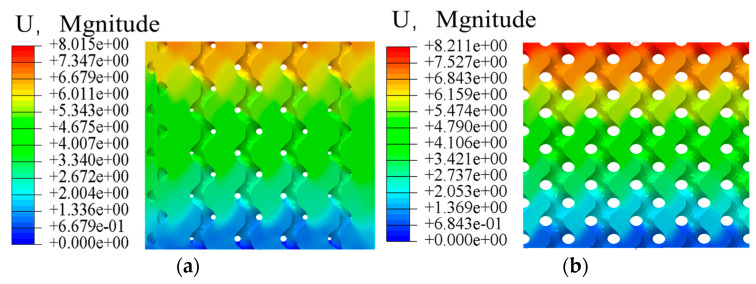
Deformation of homogeneous, gradient porous structure: (**a**) Gradient porous structure; (**b**) Homogeneous porous structure.

**Figure 5 materials-17-01137-f005:**
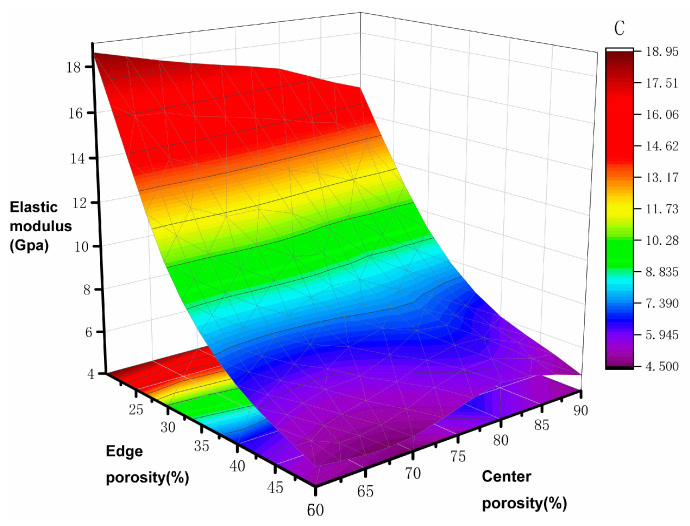
Modulus of elasticity and porosity distribution pattern.

**Figure 6 materials-17-01137-f006:**
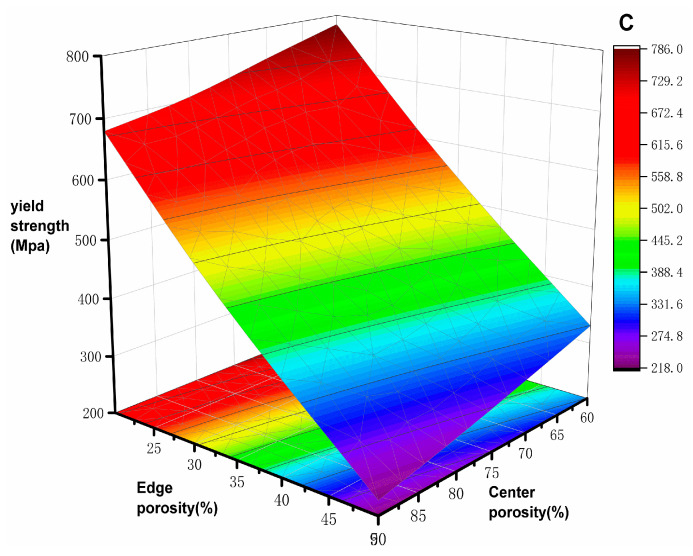
Yield strength and porosity distribution pattern.

**Figure 7 materials-17-01137-f007:**
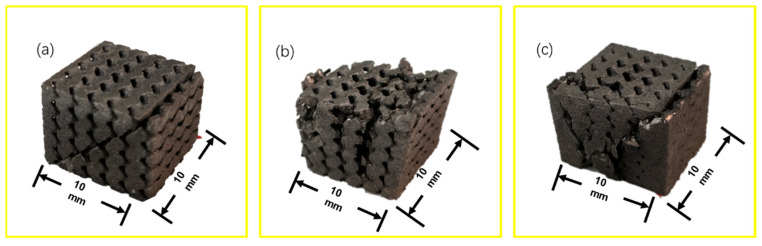
Main failure modes: (**a**) 45° type; (**b**) plumbline type; (**c**) inverted triangle.

**Figure 8 materials-17-01137-f008:**
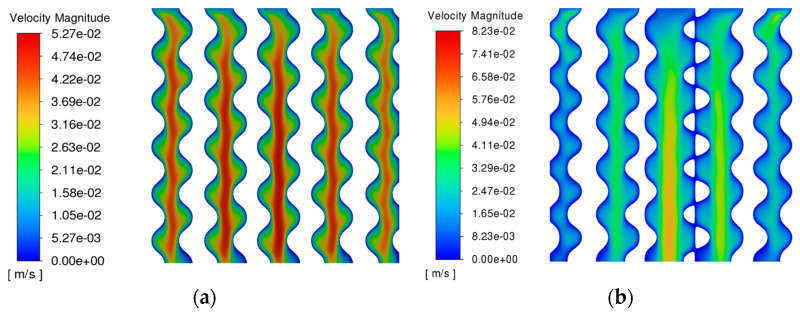
Flow rates of homogeneous and gradient porous structures: (**a**) Homogeneous porous structure; (**b**) Gradient porous structure.

**Figure 9 materials-17-01137-f009:**
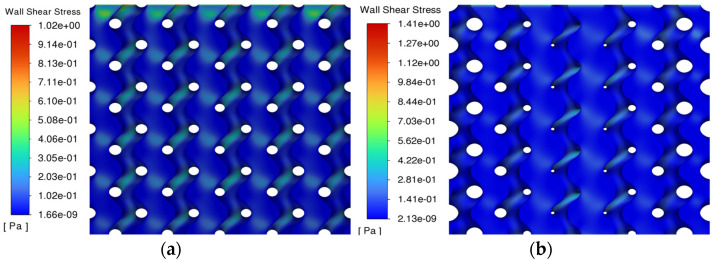
Shear stress distribution of homogeneous and gradient porous structures: (**a**) Homogeneous porous structure; (**b**) Gradient porous structure.

**Figure 10 materials-17-01137-f010:**
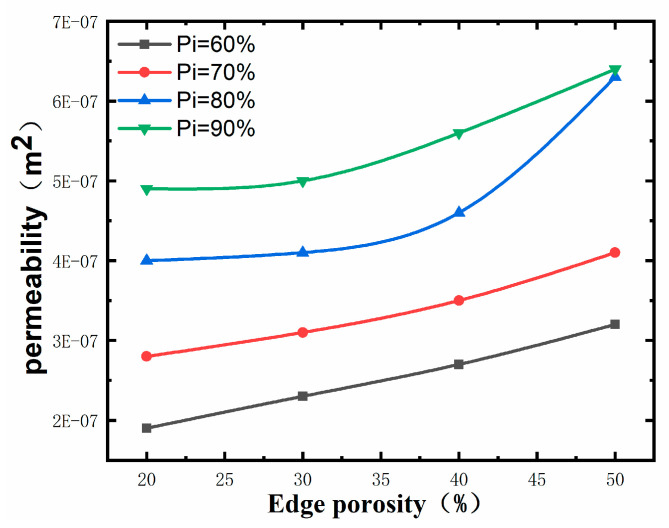
Influence pattern of edge porosity on permeability.

**Figure 11 materials-17-01137-f011:**
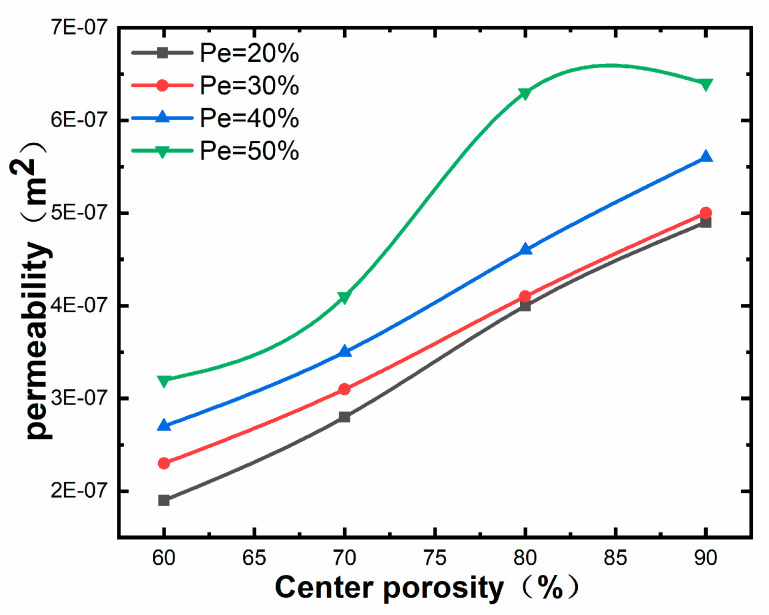
Influence law of center porosity on permeability.

**Figure 12 materials-17-01137-f012:**
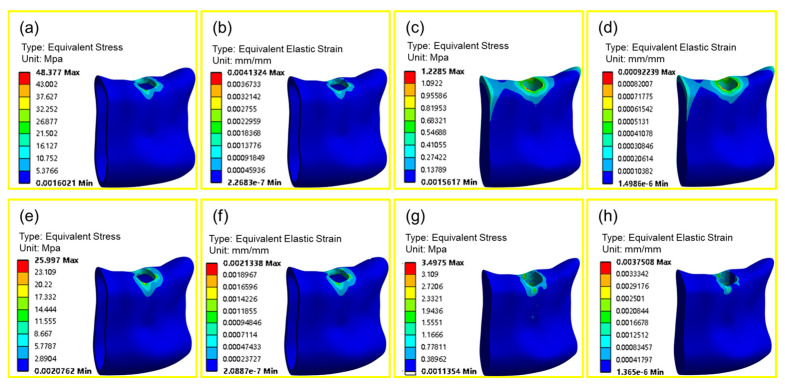
Biomechanical properties of homogeneous and gradient porous structures: (**a**) Homogeneous cortical bone stress distribution; (**b**) Homogeneous cortical bone strain distribution; (**c**) Homogeneous cancellous bone stress distribution; (**d**) Homogeneous cancellous bone strain distribution; (**e**) Gradient cortical bone stress distribution; (**f**) Gradient cortical bone strain distribute; (**g**) Gradient cancellous bone stress distribution; (**h**) Gradient cancellous bone strain distribution.

**Table 1 materials-17-01137-t001:** Printer parameter settings.

Project	Parameter
Laser power	200 W
Scanning speed	1200 m/s
Scanning distance	140 μm
Thickness of powder layer	30 μm

**Table 2 materials-17-01137-t002:** Modulus of elasticity and Poisson’s ratio of materials.

Structure	Structure Modulus of Elasticity (Gpa)	Poisson’s Ratio
Porous implants	110	0.35
Cortical bone	13.7	0.3
Cancellous bone	1.37	0.3

**Table 3 materials-17-01137-t003:** Comparison of simulated and experimental mechanical properties of gradient porous structures.

Type of Structure	Modulus of Elasticity (Experimental)	Yield Strength (Experimental)	Modulus of Elasticity (Simulation)	Yield Strength (Simulation)
G60-20	13.9 GPa	415.68 MPa	18.9 GPa	784.3 MPa
G60-30	6.6 GPa	420.59 MPa	10.1 GPa	615.5 MPa
G60-40	4.1 GPa	409.88 MPa	6.6 GPa	476.6 MPa
G60-50	3.9 GPa	175.8 MPa	5.0 GPa	331.7 MPa
G70-20	13.5 GPa	433.14 MPa	17.6 GPa	746.8 MPa
G70-30	6.3 GPa	460.24 MPa	10.6 GPa	575.3 MPa
G70-40	4.3 GPa	312.33 MPa	5.9 GPa	427.0 MPa
G70-50	3.8 GPa	191.45 MPa	4.5 GPa	293.3 MPa
G80-20	13.4 GPa	423.26 MPa	17.1 GPa	705.0 MPa
G80-30	4.4 GPa	499.79 MPa	9.0 GPa	535.5 MPa
G80-40	4.2 GPa	273.84 MPa	7.5 GPa	391.2 MPa
G80-50	3.8 GPa	138.94 MPa	5.9 GPa	219.5 MPa
G90-20	11.6 GPa	354.38 MPa	15.3 GPa	681.5 MPa
G90-30	4.8 GPa	405.76 MPa	9.1 GPa	507.2 MPa
G90-40	3.9 GPa	300.08 MPa	5.9 GPa	360.1 MPa
G90-50	3.8 GPa	176.13 MPa	4.8 GPa	230.7 MPa

**Table 4 materials-17-01137-t004:** Comparison of human porous implant performance in the literature.

Structure Type	Porosity	Modulus of Elasticity	Yield Strength	
TPMS Gyroid	62–82%	22.1 GPa	205 MPa	Bora [39]
I-WP	55%	13.96 GPa	660.6 MPa	Zeng [40]
Cubic porous	67%	20.03 GPa	205.4 MPa	Rakesh [41]
TPMS Gyroid	90–20%	15.3 GPa	681.5 MPa	This paper

## Data Availability

Data are contained within the article.

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
