# Peer review of "Parametric Design of Porous Structure and Optimal Porosity Gradient Distribution Based on Root-Shaped Implants"

_materials, 2024, doi:10.3390/ma17051137_

Round 1
Reviewer 1 Report
Comments and Suggestions for Authors
The idea for the article is very interesting, but you still need to put in some work and tidy up the text. Remove repetitive sentences, add appropriate titles to the drawings. Whether you did only simulations in your studies or also the practical part, please explain. More information below.
Introduction
Line 43
. The root-analogue implant is a root-analogue implant – you should to change this sentence, please
An interesting idea to create these types of implants. However, there is probably a problem that comes to my mind that would require clarification. Very often, before a tooth is lost, the root tissue atrophies, so after extraction the resulting pocket is very shallow and the use of such custom-made implants could result in it not having the proper retention in the event of loads.
The same as in the case of some root crown posts.
Line 70
three-periodic minimal surface- I think triple-periodic minimal surfaces (TPMS?
Line 74
demonstrated that the TPMS structure has a biological natural affinity for the TPMS structure-? Please expalin because it means the same?
Line 89
Shi et al. (2019) parametrically designed porous structures based on 3D periodic miniaturized surfaces (TPMS)- but above you said that TPMS means triply-periodic minimal surface?
At the end of the introduction part, a thesis should be stated. or aim of this study. What do you expect if you design a given structure?
Part 2 Structural design of Gyroid porous bone scaffolds
Line 113
If the accession numbers have not yet been obtained at the time of submission, please state that they will be provided during review. They must be provided prior to publication.-please explain what date are you thinking about, you looked on date base, so data has been already published?
Figure 1 should have been mentioned in the text before it was included. It would be good to include what it represents in the description, because figures and tables, even if taken from the text, should be understandable on their own.
Equation 1 and what means fg?
Line 139 and what dose P mean in equation 2?
Equation 2 what a means?
Line 177
Esen and Bor similarly recommended the use of interconnected macroporous structures with porosity more significant than 55% for dental applications to improve inward cell growth [32]but in the ref 32 is
Huang, C. C.; Li, M. J.; Tsai, P. I.; Kung, P. C.; Chen, S. Y.; Sun, J. S.; Tsou, N. T. Novel design of additive manufactured hollow porous implants. Dent Mater 2020, 36(11), 1437-1451
Line 181
paranoia function- could you give me reference for this kind of math function, please?
Mathematica software built-in RegionPlot3D function- producer country?
Mmics?- https://en.wikipedia.org/wiki/Monolithic_microwave_integrated_circuit
Line 206
SLM metal printer- producer country? What kind of materiel do you use for this printing, describe a little more such process, please?
universal testing machine, processing of the experimental data, fitting the data in the 211
linear phase, with the slope being the modulus of elasticity- please xpain this experimental part a little bit more, what kind of icompressive strength instrument do you use? Speed of crushing… Data from the tests?
Line 214
comparing the structural parameters of the porous structures in the experiments and simulations, to Compare the structural parameters of the absorbent structure in experiment and simulation- but this sentences mean the same?
Figure 2 the same as figure 1 , place the figure after the explanation in the text, describe it, please.
Workbench- producer country?
Line 225
, in order to keep the model in a laminar flow state, the inlet velocity is given as 0.01m/s,- compare with the line 229
. In order to keep the model in a laminar flow state, the inlet velocity is given as 0.01m/s
Geomagic Studio- producer country, please add?
he structure after 3D reconstruction is shown in Figure 2- which tooth in Figure 2 was scanned? Is the CT image of the entire head? so it's hard to see a single tooth there?
Line 260
A load of 100 N was applied over the crown to replace the occlusal force, and the applied load was inclined from the palatal to the buccal side at 45° to the long axis of the implant- This is already a description of the test you performed, but how were the samples prepared and how many were there? Can you add a photo/ How the samples were fixed in the crushing machine?
Figure 3 and Figure 4 unites are missing, please add.
The simulation results of elastic modulus and yield strength are shown in the table- what number?
Line 318
Table 2. Comparison of simulated and experimental mechanical properties of gradient porous
Structures- so did you perform any experiments or just simulations? Which results are obtained by simulation and which are experimental, please explain?
Figure 10 and Figure 11 Figure 12 are not mentioned in the text
Part 4 and part 5, results and discussion are in two chapters? In the normal scientific paper it should have following parts; introduction, materials and methods, results, discussion.
Good luck in further research!
Comments on the Quality of English Language
there are repeated words in the text, it would be worth having it read by a native speaker who is familiar with the topic
Reviewer 2 Report
Comments and Suggestions for Authors
The subject of the manuscript entitled “Parametric design of porous structure and optimal porosity gradient distribution based on root-shaped implants” refers to finding a material with an adequate porous structure in order to reduce the stress shielding effect and to promote a long-term stability of dental implants. For this purpose, the authors chose a TPMS-G model and by modifying the pore size and porosity of the G unit, they tried to obtain a dental implant with a better march between mechanical and biological properties. The topic of the manuscript is certainly interesting. The manuscript describes and discusses well the results and the experiments. The summary is clear and precise. However I have some suggestions:
1. The originality of the study should be highlighted in the Introduction Section.
2. The performance of the porous implants obtained in this study must be compared with those obtained for other similar materials presented in the literature.
3. The title “Results and Discussion” of Section 5 must be replaced with Conclusions.
4. Triple-periodic minimal surfaces must be written as triply-periodic minimal surfaces.
Reviewer 3 Report
Comments and Suggestions for Authors
The article entitled “Parametric design of porous structure and optimal porosity gradient distribution based on root-shaped implants” has been submitted for publication in the journal Materials (MDPI).
The research is a numerical study dealing with the design of porous root-shaped implants made of titanium alloy. The objective is to evaluate the impact of a porosity gradient on the biological and mechanical compatibility of a metallic bone implant. The porosity reduces the elastic modulus of the implant and makes it closer to that of bone tissues, then decreases the stress shielding caused by the mechanical mismatch between both materials.
The results show that controlling the porosity is an appropriate solution to improve the biomechanical compatibility of metallic implants with bones.
In my opinion, this article contains valuable results that are appropriately discussed. The study is clear, scientifically well-conducted, and undoubtedly relevant to the field of bone implants.
I think this article deserves to be published in the journal Materials (MDPI) after considering the following minor points:
- in the whole manuscript, the letter “P” must be a capital letter in the units “GPa” and “MPa”.
- In line 74, the following sentence must be checked: “…demonstrated that the TPMS structure has a biological natural affinity for the TPMS structure…”.
- In line 76, the symbol for titanium is “Ti” not “TI”.
- In line 80, “FE” must be defined.
- In line 175, the first author’s name in reference [31] is “Jetté”, not “Jett”.
- In lines 177 to 179, the reference [32] does not refer to Esen and Bor. All the references must be carefully checked.
- In line 303, what is “Yingli shielding”?
- section 5 should be named “Conclusions” instead of “Results and Discussion”. In addition, this section should contain perspectives on this work. What will this study be used for? What is the next step?
Comments on the Quality of English LanguageThe English language is good, only minor corrections are necessary.
Round 2
Reviewer 1 Report
Comments and Suggestions for Authors
All my suggestions have been added to the text. I would like to thank the authors for this. There are no further reservations and I wish you good luck in your further research and much success.